

# Stratigraphic noise and its potential drivers across the plateau of Dronning Maud Land, East Antarctica

Nora Hirsch[1], Alexandra Zuhr[1], Thomas Münch[1], Maria Hörhold[2], Johannes Freitag[2], Remi Dallmayr[2], and Thomas Laepple[1, 3]

[1]Alfred-Wegener-Institut, Helmholtz Centre for Polar and Marine Research, Potsdam, Germany
[2]Alfred-Wegener-Institut, Helmholtz Centre for Polar and Marine Research, Bremerhaven, Germany
[3]University of Bremen, MARUM – Centre for Marine Environmental Sciences and Faculty of Geosciences, Bremen, Germany

**Correspondence:** Nora Hirsch (nora.hirsch@awi.de)

**Abstract.** Stable water isotopologues of snow, firn, and ice cores provide valuable information on past climate variations. Yet single profiles are generally not suitable for robust climate reconstructions. Stratigraphic noise, introduced by the irregular deposition, wind driven erosion and redistribution of snow, impacts the utility of high resolution isotope records, especially in low accumulation areas. However, it is currently unknown how stratigraphic noise differs across the East Antarctic Plateau and

how it is affected by local environmental conditions. Here, we assess the amount and structure of stratigraphic noise at seven sites along a 120 km transect on the plateau of Dronning Maud Land, East Antarctica. Replicated oxygen isotope records of 1 m length were used to estimate signal to noise ratios as a measure of stratigraphic noise, while accumulation rates (43–64 mm w.eq. a-1), snow surface roughness and slope inclinations gave insights on the local environmental settings. While we found a high amount of stratigraphic noise at all sites, there was also a considerable amount of spatial variability. At sastrugi

dominated sites, higher stratigraphic noise coincided with higher surface roughness, steeper slopes, and lower accumulation rates, probably related to increased wind speeds. These results provide a first step to modelling stratigraphic noise and guide future exhibitions in adjusting their sampling strategies to maximise the usage of high resolution isotope records from low accumulation regions.

## 1 Introduction

The East Antarctic ice sheet is a valuable climate archive. Stable water isotopologues of snow, firn, and ice core records store information on past temperature variations (e.g., Jouzel and Masson-Delmotte, 2010; Dansgaard, 1964), especially on past glacial and interglacial periods (EPICA community members, 2004, 2006). However, the interpretation of isotope records at subannual to decadal resolutions is hampered by significant uncertainties (e.g., Casado et al., 2020; Laepple et al., 2018; Münch and Laepple, 2018) which limit the usability of this archive to quantify recent global warming impacts on the Antarctic (Stenni

et al., 2017; Jones et al., 2016). Several processes, such as precipitation intermittency (Laepple et al., 2011; Helsen et al., 2005), water vapour exchange and sublimation (e.g., Wahl et al., 2021; Town et al., 2008), and isotopic diffusion (e.g., Laepple et al., 2018; van der Wel et al., 2015; Johnsen et al., 2000) introduce noise to the isotopic temperature imprint in snow, firn, and ice



cores. Additionally, accumulation does not occur as perfectly stratified layers but is affected by irregular depositions as well as aeolian erosion and redistribution (Zuhr et al., 2021; Picard et al., 2019). These processes introduce a non-climatic variability in the isotope profiles, known as stratigraphic noise (Fisher et al., 1985), which reduces the usability of single isotope profiles for climate reconstructions as they are not representative (e.g., Münch et al., 2016).

While atmospheric circulation, temperature, and precipitation intermittency introduce differences in the isotopic compositions on scales of hundreds of kilometres (Goursaud et al., 2018; Münch and Laepple, 2018), stratigraphic noise has been defined as the uncorrelated part between two or more isotope profiles at local scales (Fisher et al., 1985). Furthermore, Münch et al. (2016) found the decorrelation length of stratigraphic noise to be around 5 to 10 m on the plateau of Dronning Maud Land (DML), which means that the stratigraphic noise in one snow core is independent from the noise of an adjacent profile at a distance of more than 5-10 m.

Stratigraphic noise hampers the extraction and interpretation of climate signals, especially on subannual to decadal scales where accumulation rates are low (e.g., Jones et al., 2014; Karlöf et al., 2006; McMorrow et al., 2002; Sommer et al., 2000). For instance, with accumulation rates in Greenland of 240-600 mm w.eq. $a^{-1}$, it is possible to extract seasonal and annual signals from the isotopic imprints in firn cores (Vinther et al., 2010). However, with low accumulation rates of 40 to 90 mm w.eq. $a^{-1}$ in DML (Oerter et al., 2000), seasonal snow layers can be completely eroded (Helsen et al., 2005). Accordingly, discrepancies between isotope profiles and observed temperatures are smaller in coastal areas of East Antarctica compared to the drier inland plateau of DML (Helsen et al., 2005). This is further evidenced by the signal to noise ratios (SNRs), which measure the ratio of the common (spatially coherent) isotopic signal over the independent stratigraphic noise: while Fisher et al. (1985) found SNRs between 1.1 and 2.7 (at 140–520 mm w.eq $a^{-1}$) for annually resolved ice core records from Greenland, Graf et al. (2002) obtained a SNR of only 0.14 on the plateau of DML.

Although several studies have identified stratigraphic noise as a crucial limiting factor for high resolution ice core signal interpretation, it remains unknown how stratigraphic noise differs spatially, e.g., across the East Antarctic Plateau (EAP), and how it is related to local environmental properties like accumulation rate, slope inclination, and surface roughness. Knowledge of these relationships would allow to optimise the selection of sampling sites for extracting snow and firn cores for the purpose of high-resolution climate reconstructions. Furthermore, this knowledge would enable stratigraphic noise to be simulated in proxy system models (Casado et al., 2018; Dolman and Laepple, 2018; Dee et al., 2015). An improved quantitative understanding would also result in more accurate estimates of past climate variability as it would allow us to correct for stratigraphic noise within the spectral domain (e.g., Münch and Laepple, 2018; Laepple et al., 2017). It might further enable the use of replicate cores as a proxy for past surface roughnesses and related wind speeds, as suggested by Barnes et al. (2006).

In this study, we use SNRs to quantify stratigraphic noise in high resolution isotope records collected from seven sites in DML (EAP). We relate differences in stratigraphic noise to varying local environmental properties such as slope inclination, surface roughness, and accumulation rate in order to identify potential underlying environmental drivers.





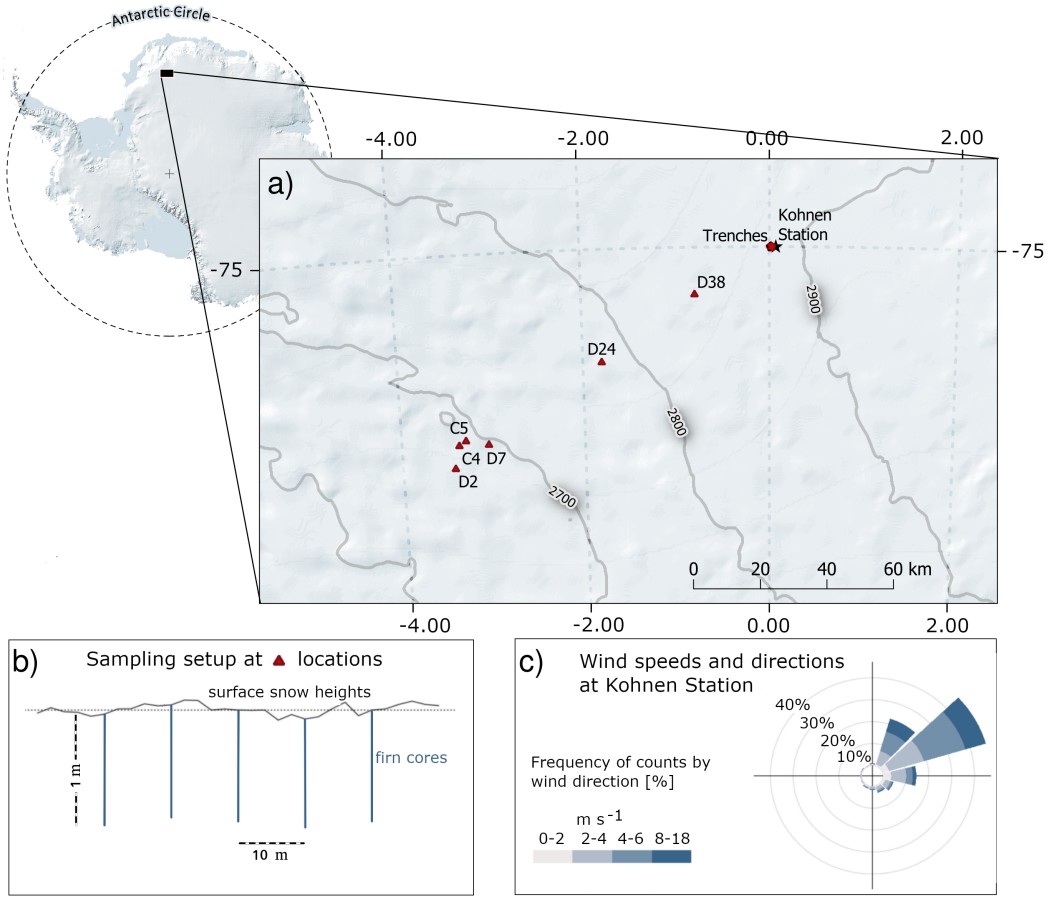

**Figure 1.** a) Study area on the plateau of Dronning Maud Land (DML), East Antarctica, with sampling sites D2, C4, C5, D7, D24, C38, and Kohnen trenches (Münch et al., 2016, 2017), shading represent the surface elevation above sea level (RAMP2 Liu et al., 2015). b) Example schematic illustrating the collection of replicate core samples at each sampling site. c) Wind speeds and direction at Kohnen Station (AWS 9, 2012-2018).

## 2  Materials and methods

### 2.1  Study area

The sampling sites are situated on the EAP along a 120 km transect that rises gently from 2685 to 2892 m.a.s.l. near Kohnen Station (Wesche et al., 2016, 75.002°S, 0.007°W, 2892 m.a.s.l.) (Fig. 1a). Between 2012 and 2018, the annual mean temperature at Kohnen station was -40.9 °C and the annual mean wind speed was 4.31 m s$^{-1}$ (AWS 9, Reijmer and van den Broeke, 2003). The region is characterized by katabatic winds consistently blowing downslope (e.g., Broeke and Lipzig, 2003; Parish and Cassano, 2003) from north-easterly directions (Fig. 1c), with wind speeds in excess of 10 m/s occurring about 10-20 times per





year (Birnbaum et al., 2010; Reijmer and van den Broeke, 2003). Only few precipitation events, introduced by low pressure systems, bring most of the annual precipitation amount (Schlosser et al., 2010; Reijmer and van den Broeke, 2003; Noone et al., 1999). Average accumulation rates over the past 200 years have been ∼60 mm w.eq. (Oerter et al., 2000), possibly with

a considerable increase during the last 100 years (Medley and Thomas, 2019). The snow surface is mainly comprised of small dunes and sastrugi with horizontal scales of the order of a few metres (Birnbaum et al., 2010).

## 2.2   Snow core sampling

A set of snow profiles was sampled in December 2018 at various sites along the ∼120 km transect south-west of Kohnen Station (Fig. 1a). At each of the six sampling sites (D2, C4, C5, D7, D24 and D38) we extracted five 1 m snow cores along a

line running perpendicularly to the dominant large scale wind direction, with a 10 m interprofile spacing (Fig. 1b). Each snow profile was extracted by vertically inserting a 1 m carbon fibre pipe into the sidewall of a snow-pit. At Kohnen Station, the collected snow profiles were cut horizontally into 1.1 cm (the upper 16.5 cm) and 3.3 cm (the lower part) slices, accounting for the diffusion length of ∼3 cm at 1 m depth (Laepple et al., 2018). Due to small compression and expansion errors during handling, transporting, and cutting, this resulted in 41 to 43 samples per profile, which led us to assume a maximum depth

uncertainty of 2 cm. All cut snow samples (N = 1249) were packed in plastic bags and transported in a frozen state to Germany for further analysis.

## 2.3   Stable water isotope measurements

The stable water isotopic composition ($\delta^{18}$O, $\delta$D) of the snow samples was measured using a Cavity Ring-Down Spectroscopy instrument (CRDS) of PICARRO Inc. (model L2140-i) in the Stable Isotope Facility at the Alfred Wegener Institute in Potsdam,

Germany, applying post-run corrections as described in Münch et al. (2016). Scaling to the VSMOW/SLAP (Vienna Standard Mean Ocean Water/Standard Light Antarctic Precipitation) scale results in the $\delta$-notation which describes the ratio of heavy to light isotopes in ‰. In-house standards were used for quality control. The mean combined measurement uncertainty is 0.07 ‰ for $\delta^{18}$O and 0.5 ‰ for $\delta$D (root-mean-square deviation, RMSD). In the following we focus on the $\delta^{18}$O values.

## 2.4   Trench isotope subset

To complement the datasets from the six sites along the transect, we use already published $\delta^{18}$O profiles from Kohnen Station (Fig. 1a) derived in the years 2012/13 and 2014/15 by Münch et al. (2016, 2017). Four snow trenches, in the following called Kohnen trenches, were excavated by a snow blower perpendicularly to the local snow dune direction. Snow profile samples were collected off the trench walls, resulting in snow profiles of high vertical as well as horizontal resolutions (Table 1). For the comparison of the trench data to the new datasets, we divided the trench data into 10 subgroups, each composed of four to

five one metre deep $\delta^{18}$O profiles at distances of 10 ± 1 m from one another (Table 1).



**Table 1.** Summary of the datasets which are used within this study. Subgroups of the Kohnen trenches, described in Sect. 2.4, are used to compare the resulting signal to noise ratios (SNRs) to the ones of locations D2, C4, C5, D7, D24 and D38.

| Site | Sampling time | Spacing [m] | Depth [m] | Resolution [cm] | number of profiles | SNR subgroups |
|---|---|---|---|---|---|---|
| D2, C4, C5, D7, D24, D38 | Dec. 2018 | 10 | 1 | 1.1 - 3.3 | 5 each | - |
| Trench t13.1[a] | 2012/13 | 0.1 - 2.5 | 1.2 | 3 | 38 | 5 |
| Trench t13.2[a] | 2012/13 | ∼ 10-20 | 1.2 | 3 | 4 | 1 |
| Trench t15.1[b] | 2014/15 | 5 | 3.4 | 3 | 11 | 2 |
| Trench t15.2[b] | 2014/15 | 5 | 3.4 | 3 | 11 | 2 |

*a* Münch et al. (2016), *b* Münch et al. (2017)

## 2.5 Definition and quantification of stratigraphic noise

We define stratigraphic noise as the fraction of the isotope records that differs between adjacent profiles, while the signal is the isotope variations that those profiles have in common. The relative ratio of the common (signal variance) to the independent (noise variance) portion in the given data is the so-called signal to noise ratio (SNR). This ratio can be derived based on the pairwise correlation coefficient, $r_{XY}$, between two isotope profiles X and Y, with

$$SNR_{XY} = \frac{r_{XY}}{1 - r_{XY}} \tag{1}$$

(Fisher et al., 1985). In the absence of any noise, we would expect $r_{XY}$ = 1, i.e., a perfectly stratified isotopic imprint. As the amount of stratigraphic noise increases, the pairwise correlations and hence the SNRs will decrease. SNR thus provides a quantitative measure allowing us to objectively determine the proportion of stratigraphic noise in adjacent (intrasite) isotope records and to make intersite comparisons. At each new sampling site, SNRs were estimated based on the average pairwise correlation coefficient (n = 10) between the five $\delta^{18}$O profiles (interpolated to a 0.1 mm resolution) at their absolute height reference (Appendix A). The SNR of the Kohnen trenches was determined as the mean of all pairwise correlations of all subgroups (n = 95). Due to statistical uncertainty, SNR estimates can be negative; as this has no physical meaning, SNR estimates of < 0 are set to 0. In addition, we employ bootstrap resampling to further assess the uncertainty of the SNR estimates. With this approach, we resample the pairwise correlation coefficients with replacement, calculate the SNR from the resamples, and derive the confidence intervals from the distribution of SNRs.

## 2.6 Environmental properties

The amount of stratigraphic noise in the isotope records is compared to the local slope inclination, accumulation rate, and surface roughness.

Slope inclinations are hereby defined as the local inclination along 10 km down the transect (azimuth of ∼57.1°). They were derived using 200 m resolution data from the REMA digital elevation model (DEM, Howat et al., 2019) and vary between 0.3 and 3.1 m km$^{-1}$.





We use the average accumulation rates over the last 200 years as determined by Rotschky et al. (2004) for the same transect using ice-penetrating radar. This data showed that accumulation rates at sites D2, D7, D24, and D38 varied between ~43 and ~59 mm w.eq. a$^{-1}$. At Kohnen Station, the corresponding value is ~64 mm w.eq. a$^{-1}$ (EPICA community members, 2006).

Snow heights were measured with a 2 m horizontal resolution and a height accuracy of ±1 cm along the line of the five snow cores at each site (60 m length; cf., Fig. 1b) using a Geodät levelling device. Surface roughness is defined by the standard deviation SD$_{sh}$ of these surface heights. To assess past snow surface heights and roughnesses, common isotopic extremes, i.e., isochrones, were assigned and traced wherever possible.

## 3 Results

### 3.1 $\delta^{18}$O profiles

Mean $\delta^{18}$O values at the new sampling sites range from -43.9 ‰ to -44.3 ‰ (Table 2, Fig. 2) with no significant differences between sites (two-sided Student t-Test; p > 0.05). The mean vertical variance within the arrays of the isotope profiles is $\sigma^2$ = 8.9 ‰$^2$ (SD 2.9 ‰$^2$) and thus slightly higher than the mean horizontal variance of $\sigma^2$ = 7.5 ‰$^2$ (SD 5.5 ‰$^2$). Between 4 and 9 local $\delta^{18}$O maxima were observed in the isotope profiles (Fig. 2) with a mean peak to peak amplitude of 5.0 ‰ (SD 3.0 ‰).

### 3.2 Recent and past snow surfaces and surface roughnesses

The surface roughness, defined by the standard deviation SD$_{sh}$ of the surface heights, varies between 3.5 cm (Kohnen trenches) and 8.6 cm (D24) (Table 2). The variations at the Kohnen trench site are significantly smaller compared to the other sites (two-sided f-Test, p < 0.05). Sites D24, D7, and C4 as well as sites D2, C5, and D38 form two distinct clusters, with no significant intracluster differences in surface roughness, but considerable intercluster variations.

At two sites (D2 and D38) it was possible to tentatively trace past snow surface variations by manually tracking local isotope extremes, i.e., isochrones (Fig. 2). This could also be done, with a high degree of confidence, at two of the Kohnen trenches (13.1 and 13.2; Table 1) by Münch et al. (2016). In contrast, the assignation of common peaks was more ambiguous and uncertain at site D38 and particularly at D2 where one to two cycles might have been missed in the top 25 cm. At all three sites, the isochrones (black horizontal lines in Fig. 2) exhibit a similar degree of roughness as the snow surface (grey dashed lines in Fig. 2) with SD$_{surface}$ = 3 cm and SD$_{past}$ = 3.7 cm at the Kohnen trenches (Münch et al., 2016), SD$_{surface}$ = 3.7 cm and SD$_{past}$ = 3.5 cm at D2, and SD$_{surface}$ = 4 cm, SD$_{past}$ = 3.4 cm at D38. At D2, certain surface features may have been preserved over several years resulting in correlation coefficients between isochrone profiles and the surface heights (grey dashed lines, Fig. 2) of between 0.47 and 0.87. At site D38, the same test results in correlation coefficients between -0.79 and 0.90, indicating an annual reorganisation of the stratigraphy which are consistent with earlier findings at the Kohnen trenches (Münch et al., 2016).

At site D24, equally strong isotopic values were found at depths of about -20 and -60 cm. Depending on the exact depth of the traced isochrones, the resulting SD$_{past}$ varies between 1 and 2.8 cm, which differs significantly from the SD$_{surface}$ of 10





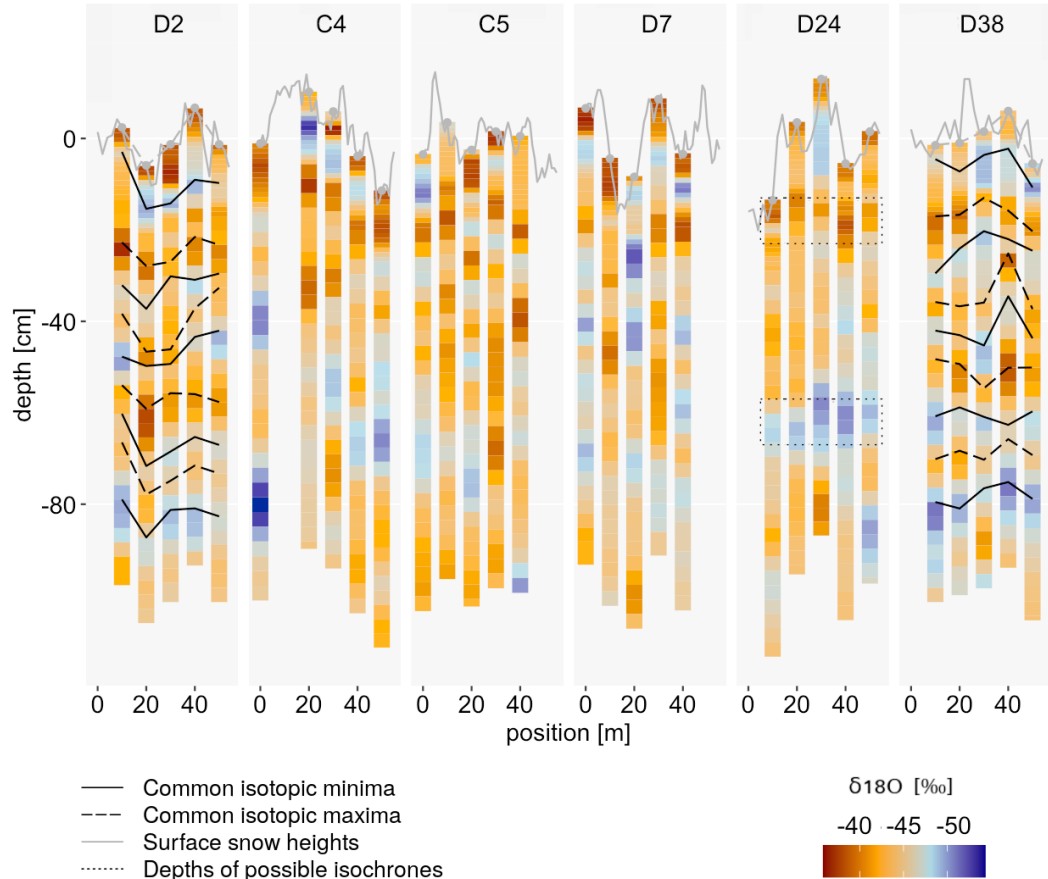

**Figure 2.** $\delta^{18}$O profiles at each of the new sampling sites plotted against snow depth. The colour scale shows the $\delta^{18}$O values. Black lines trace possible common isotope minima (solid lines) and maxima (dashed lines), wherever it was possible to identify common peaks (sites D2 and D38). Grey lines indicate the surface topography at 2 m (solid line) and 10 m (dashed line) horizontal resolutions. The black dashed boxes at site D24 indicate the depths of similar isotopic peaks, even if consecutive isochrones could not be assigned.

cm. However, as for sites C4, C5, and D7, consecutive $\delta^{18}$O isochrones at D24 could not be traced with sufficient confidence

due to strong irregularities in the isotopic cycles.

### 3.3   Signal to noise ratios

The SNRs provide an estimate of stratigraphic noise imprinted in the $\delta^{18}$O profiles (Fig. 3 and Table 2). We estimated them for each of the newly sampled sites and for the Kohnen trenches. Overall, the SNRs range from 0 (C5) to 0.77 (Kohnen trenches and D24). The statistical uncertainty of SNR estimates is lower at sites with low SNRs (C4, C5, D7) as well as at the Kohnen trenches, which is due to the higher amount of $\delta^{18}$O profiles available. The highest uncertainty was estimated for sites D38 and

D24. Considering these uncertainties, the SNR at the Kohnen trenches is statistically significantly elevated (p < 0.05) compared




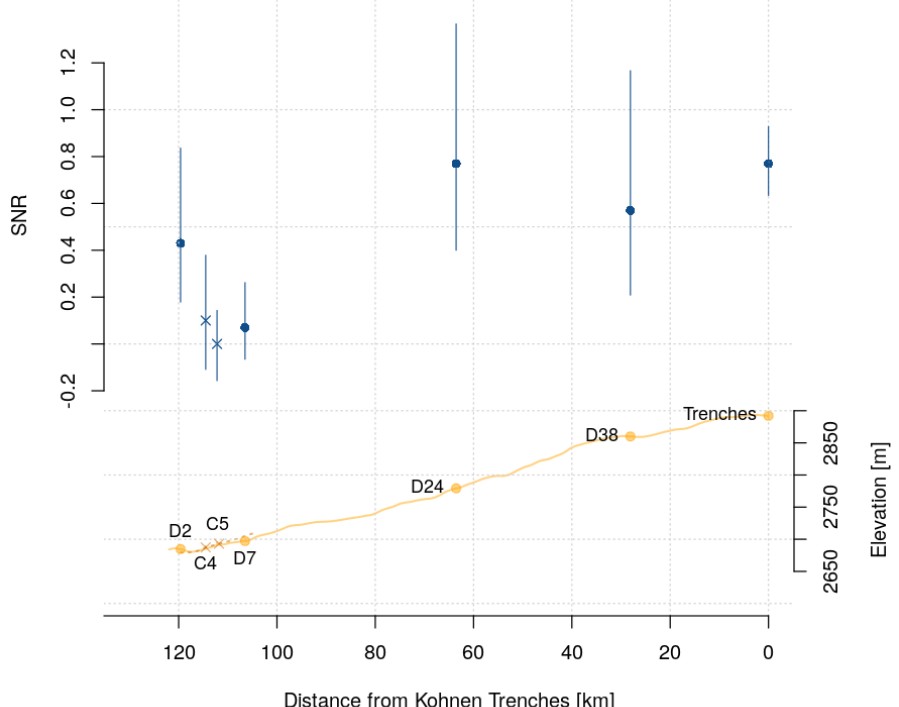

**Figure 3.** Signal to noise ratio (SNR) and the 95% confidence intervals (blue lines) at the different sites (top panel) and their positions along the transect (bottom panel) with elevations from the RAMP2 DEM (yellow line, Liu et al., 2015). The elevation data at C4 and C5 was collected along a transect that was slightly (several km) offset to the north.

to C4, C5 and D7. Furthermore, the SNR at D24 is significantly higher than at C5 and D7 while the SNR at D2 is significantly higher than at C5.

# 4 Discussion

We found different isotopic structures and different amounts of stratigraphic noise within replicate 1 m isotope records from seven different sites across a 120 km transect, coupled with differing snow surface features, slope inclinations, accumulation rates and surface roughnesses. In this section, we discuss the observed differences and possible relationships, and conclude some recommendations for future studies.

## 4.1 Isotope profiles and snow height evolution related to sastrugi and glazed surfaces

In shallow isotope profiles, which are not yet superimposed by diffusion (Laepple et al., 2018), $\delta^{18}O$ minima represent snow that fell at colder atmospheric temperatures, i.e., in austral winter, while the maxima represent warmer temperatures from sum-





**Table 2.** Summary of the isotopic and environmental parameters for all sampling sites showing $\delta^{18}$O [‰] and their standard deviations SD$_{\delta^{18}O}$, signal to noise ratios (SNRs), mean accumulation rates (A) derived from ground penetrating radar (Rotschky et al., 2004) in water equivalent [mm w.eq. a$^{-1}$] and snow equivalent [cm snow a$^{-1}$], surface roughness (SD$_{SH}$ [cm]), the maximum height difference between two local/adjacent snow cores [cm] and slope inclinations [m km$^{-1}$]. For converting accumulation rates from water equivalent to snow equivalent we assumed a snow density of 344 kg/m$^3$, which is the overall arithmetic mean for all sites (n = 19).

| Site | $\delta^{18}O$ [‰] | SD$_{\delta^{18}O}$ [‰] | SNR | A [mm w.eq. a$^{-1}$] | A [cm snow a$-1$] | SD$_{SH}$ [cm] | Max. height difference [cm] | Slope [m km$^{-1}$] |
|------|------|------|------|------|------|------|------|------|
| D2 | -44.1 | 3.0 | 0.43 | 58.7 | 20.2 | 4.75 | 12.6 | 0.55 |
| C4 | -44.6 | 3.3 | 0.10 | - | - | 7.23 | 11.3 | 1.84 |
| C5 | -43.5 | 2.7 | 0 | - | - | 5.50 | 7.0 | 2.22 |
| D7 | -44.6 | 2.9 | 0.07 | 43.6 | 15.0 | 7.35 | 17.0 | 1.97 |
| D24 | -44.7 | 2.7 | 0.77 | 43.3 | 14.9 | 8.55 | 26.5 | 3.12 |
| D38 | -45.2 | 2.9 | 0.57 | 52.8 | 18.1 | 5.36 | 7.5 | 0.35 |
| Trenches | -44.7 | 3.1 | 0.77 | 64.0 | 22.0 | 3.47 | 15.5 | 0.59 |

mer (e.g., Stenni et al., 2016; Dansgaard, 1964). Common isotope peaks are therefore assumed to represent snow accumulated during the same season or even during the same accumulation event. Based on the accumulation rate estimates by Rotschky et al. (2004), we expect the 1 metre profiles to have accumulated within ∼4.5 to ∼6.7 years, and to exhibit an equal number of
isotopic maxima and minima if from the same site. However, the absolute count of isotopic peaks varied considerably between profiles. Furthermore, the isotope profiles exhibited strongly varying cycle lengths and amplitudes which made it difficult to assign common isotopic peaks and which suggests a considerable redistribution and irregular accumulation. This is further confirmed by the analysis of the evolution of snow heights at D2 and D38. Particularly at D38 a refilling of troughs combined with a lower accumulation at elevated parts show typical processes of snow deposition (Zuhr et al., 2021). At D2, D38 and the
Kohnen trenches, the surface snow heights (grey solid lines in Fig. 2) show similar variations as past snow heights, indicating similar surface roughnesses over time. Furthermore, these variations indicate the presence of pronounced snow features such as dunes (Birnbaum et al., 2010).

The highest surface roughness was observed at site D24, where isotopic peaks were too variable to assign consecutive isochrones. Still, similar isotopic values indicate a very low surface roughness at depths of around -20 and -60 cm. These
nearly flat past surfaces are consistent with observed snow surface features around the site, namely large sastrugi and dunes mixed with glazed surfaces (Fig. 4a). The latter are characterised by flat and very dense snow, permeated by "thermal" cracks (Fig. 4b and c, e.g., Furukawa et al., 1996). Glazed surfaces were found to occur when high wind speeds coincide with a hiatus in accumulation, e.g., on the steep slopes in the katabatic wind region (Scambos et al., 2012; Furukawa et al., 1996) and could already be detected via satellite images within ∼200 km to the south, south-east, and north-west of the transect (Rotschky
et al., 2006).




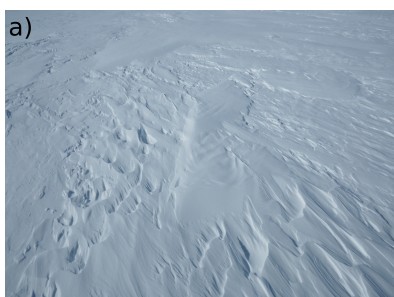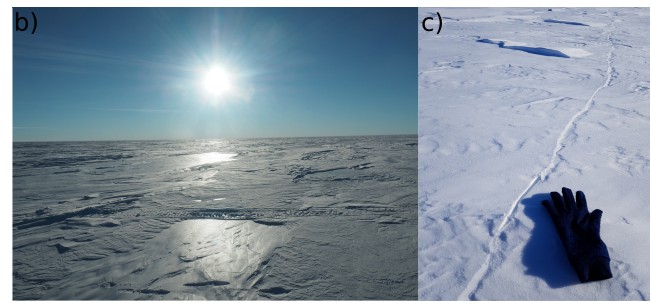

**Figure 4.** The snow surface at sites a) C4 showing the typical sastrugi, b) D24 showing a mix of sastrugi and glazed surfaces, and c) D24 showing a "thermal" crack with a glove placed next to it for scale.

## 4.2 Amount and structure of stratigraphic noise

In the newly collected isotope profiles we observed a low mean pairwise correlation coefficient of r = 0.19 (SD 0.31) which is already an indication of strong stratigraphic redistribution and irregular deposition. The pairwise correlations were furthermore independent of the spatial distance between profiles (Appendix C), which confirms that the decorrelation length of stratigraphic noise in the study area is <10 m as proposed by Münch et al. (2016) for Kohnen Station. The SNRs of < 1 indicate that stratigraphic noise explains more than 50% of the $\delta^{18}$O variability in the uppermost 1 m at all sites, which is consistent with previous findings for this region and confirms the low representativity of single isotope profiles (Münch et al., 2016, 2017; Graf et al., 2002). Along the 120 km transect, SNRs varied strongly between 0 and 0.77 which emphasises the importance of understanding its drivers.

## 4.3 Relations between stratigraphic noise and environmental properties

Despite the low number of data points, we could formulate certain hypotheses regarding the origin of stratigraphic noise. We could test whether the amount of stratigraphic noise is related to local environmental characteristics such as local accumulation rates, slope inclination, and surface roughness (Fig. 5). We exclude spatial variations in precipitation or isotopic amplitudes as factors potentially affecting SNRs (Fisher et al., 1985) as we do not expect these features to differ strongly across the spatial scale considered here (Goursaud et al., 2018; Münch and Laepple, 2018, in Appendix).

At the sites dominated by sastrugi (all except D24; cf., blue points, Fig. 5), we found that SNRs correlated with accumulation rates (r = 0.89), and was anticorrelated with surface roughness (r=-0.81, p = 0.05) and slope inclination (r=-0.92, p < 0.05). Concomitantly, isotope records at coastal sites in East Antarctica with higher accumulation rates contained a more consistent climate signal than those at lower accumulation sites in DML (Helsen et al., 2005) or near Dome Fuji (Hoshina et al., 2014). The negative correlation between SNR and surface roughness is consistent with the expectations that snow height variations did not change over the past 200 years as indicated by the isochrones at the Kohnen trenches, D2, and D38. Hence surface roughness could be regarded as an indirect measure of stratigraphic noise. In contrast to Fisher et al. (1985), who proposed that sastrugi height would be inversely proportional to accumulation rate in Greenland, we found the ratio between surface



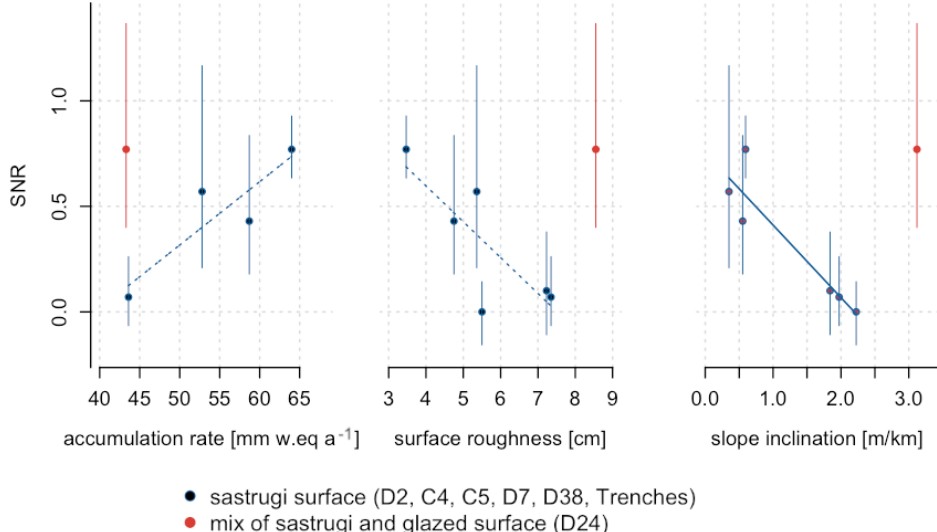

- ● sastrugi surface (D2, C4, C5, D7, D38, Trenches)
- ● mix of sastrugi and glazed surface (D24)

**Figure 5.** Comparison of signal to noise ratios (SNR) to accumulation rate A [mm w.e. $a^{-1}$], surface roughness $SD_{sh}$ [cm], and slope inclination [m km$^{-1}$]. Most sites were dominated by sastrugi while at D24 we observed a mix of sastrugi and glazed surfaces. The latter was therefore excluded when calculating a simple liner regression (dashed lines). Vertical lines indicate the 95% confidence intervals of the SNR estimates.

roughness and accumulation rate to be positively correlated to the amount of stratigraphic noise (r = 0.89). The fact that SNRs
205  are lower at sastrugi sites with higher slope inclinations is independent of the scale over which the inclination is calculated
(Appendix B).

At D24, past and present surface roughness differed from one another which could be related to the surface consisting of
a mix of sastrugi and glazed surfaces, which can alternate both spatially and temporally. At this site, the lowest accumulation
rate coincided with the highest slope inclination and highest surface roughness which yielded the highest SNR of all sites. The
210  relationships between SNR and environmental properties, as proposed for sites dominated by sastrugi, do not seem to hold up at
such a site of such extreme environmental properties and the related occurrence of glazed surfaces (e.g., Frezzotti et al., 2002;
Furukawa et al., 1996). The higher interprofile correlations at D24 appear to be driven mostly by the isotopic values associated
with the possible isochrones at depths of about -20 and -60 cm, while at intermediate depths the past surface roughnesses could
have been similarly high as the present surface roughness. As a result, the here derived isotope profiles do not exhibit annual
215  cycles. We assume that the signal, e.i., the spatially coherent part, as defined in the SNRs, does in this case not represent a
climatic signal and that the isotope profiles do therefore not provide any useful information for climate signal interpretations.




## 4.4 Relations between slope inclination, surface roughness, and accumulation rates

We also found a strong covariation among the different environmental properties. For instance, surface roughness was higher at sites with higher slope inclinations (r = 0.82, p < 0.05) and at sites with smaller accumulation rates (r = -0.97, p < 0.05), while the latter negatively correlates with slope inclinations (r = -0.82). Although Studinger et al. (2020) have questioned the validity of such relationships for large spatial scales, these close links are frequently confirmed at smaller scales: for example, accumulation rates tend to be lower in areas with steeper slopes as was confirmed in various EAP locations (e.g., Dattler et al., 2019), often associated with the underlying bedrock topography (e.g., Fujita et al., 2011; Arcone et al., 2005; Eisen et al., 2005; Black and Budd, 1964) or wind driven sublimation (e.g., Frezzotti et al., 2004) and redistribution (King et al., 2004). Steeper slopes are associated with stronger wind speeds (e.g., Dattler et al., 2019; Parish and Cassano, 2003; Broeke and Lipzig, 2003; Whillans, 1975; Endo and Fujiwara, 1973) which in turn affect snow surface features (Furukawa et al., 1996; Whillans, 1975, e.g.,). High wind speeds increase both the propagation speed and height of dunes and thereby increase surface roughness (Filhol and Sturm, 2015; Birnbaum et al., 2010; Endo and Fujiwara, 1973), at least within the physical constraints of maximum snow heights (e.g., about 1.5 m for sastrugi, Filhol and Sturm, 2015).

## 4.5 Implications for future studies

Based on the presented dataset, we conclude that the assessed environmental properties affect the amount of stratigraphic noise. However, due the low number of sampling sites, replicates, and shallow sampling depths the SNR estimates are subject to a considerable uncertainty which renders our results somewhat speculative, even if they are supported by previous studies. Furthermore, considering the significant covariance among the different environmental properties makes it difficult to disentangle their individual contributions. Additional studies, ideally collecting more replicates and deeper profiles that cover a wider range of depositional conditions, are needed to test and refine the proposed relationships. Given the significant cost and work associated with collecting samples in situ, it would be useful to test whether high frequency ground penetrating radar (e.g., Studinger et al., 2020; Rotschky et al., 2006) could serve to estimate stratigraphic noise. This would have the added advantage that it could cover much larger spatial scales which would provide information on possible large scale drivers such as precipitation patterns (Fisher et al., 1985). Moreover, surface snow accumulation studies following the approaches of Picard et al. (2019) and Zuhr et al. (2021) could provide more detailed knowledge about variations in snow stratigraphy related to depositional conditions including wind speeds.

Based on the fact that stratigraphic noise varied considerably over relatively small spatial scales (∼120 km), making a thorough decision on the drilling locations can result in a strong improvement of the isotopic signal within snow or firn cores. In general, areas with reduced slope inclinations, lower surface roughness, and higher accumulation rates are better suited to derive a high resolution climate signal. While data on small scale accumulation rates (Rotschky et al., 2004) and surface roughnesses (e.g., Studinger et al., 2020) are scarce, slope inclination data are available at high spatial resolutions for the entire Antarctic (e.g., Howat et al., 2019) and can easily be used for site selection. We assume that areas characterised by glazed surfaces are poorly suited sampling locations, as they don't exhibit a strong climate signal. Such locations can be excluded by





in situ observations, by employing remote sensing approaches to snow surface classification (Scambos et al., 2012; Rotschky et al., 2006) or selecting locations with low slope inclinations and a certain amount of accumulation (e.g., Furukawa et al., 1996).

Considering that the ice sheet topography is proposed to have been fairly constant over large parts of the EAP during the past millennium, related to the bedrock topography (Eisen et al., 2005; Steinhage et al., 1999), we can hypothesise that the amount

of stratigraphic noise imprinted in the snow over the past decade is representative for past centuries. This could be tested by collecting and analysing long (e.g., 100 m) high resolution firn cores. Furthermore, future studies should account for the fact that ice properties such as stratigraphic noise are advected horizontally due to ice flow (Arcone et al., 2005; Steinhage et al., 1999) by checking for similarly suitable environmental properties further upstream. The size of that area can be determined based on ice flow velocity (e.g., Rignot et al., 2019; Arthern et al., 2015) and desired snow or firn core length. With such a

sampling strategy, the results presented here can serve to derive a less noisy isotope signal from firn cores of probably hundreds of metres of length.

## 5   Conclusions

In this work, we assessed stratigraphic noise and its spatial variations along a transect extending for about 120 km to the south-west of Kohnen Station, Dronning Maud Land, East Antarctic Plateau. We analysed the local, non climatic variability in

$\delta^{18}$O composition at high vertical resolutions across spatial scales ranging from local (60 m) to regional ( 120 km), assessing its dependency on local environmental properties like accumulation rate, surface roughness, and slope inclination. Within the study area, we found that stratigraphic noise dominated the seasonal to interannual isotopic signal, with high spatial variations. At sites that were dominated by sastrugi, stratigraphic noise was lower if the terrain was flatter, the surface less rough, and accumulation rates higher. All these environmental characteristics are typically associated with lower wind speeds. The

combination of them is likely to make a site suitable for collecting isotope profiles from which meaningful climate signals can be extracted. Considering that the proposed relationships seem to be stationary over time, these findings could be applied to snow, firn, and ice cores that are several hundred metres in length, and thus greatly expand the usage of high resolution isotope records from the East Antarctic Plateau.

*Data availability.*   All measurements will be made available in the PANGAEA repository upon publication of this article.

**Appendix A: Height reference**

When assuming the snow surface to stay similar regarding its height variations, i.e., dunes structures, during the time of snow deposition, snow samples are processed at relative depth to each other, which is done e.g., for surface snow samples and shallow cores (e.g., Casado et al., 2016; Steen-Larsen et al., 2014). If the surface height variations are expected to have



changed throughout the time of accumulation, snow cores are instead processed at absolute heights, which was decided for the
Kohnen trenches (Münch et al., 2016, 2017). We tested this method for the new dataset, by calculating pairwise correlation
coefficients for all possible pairs of isotope profiles at each site (Fig. A1), using both the absolute and relative heights. We
would expect the correlation to be stronger with more common signal located at a similar depth, either at absolute or relative
heights. We obtained a mean correlation coefficient of r = 0.19 (SD 0.31) when using absolute heights and 0.17 (SD 0.32) for

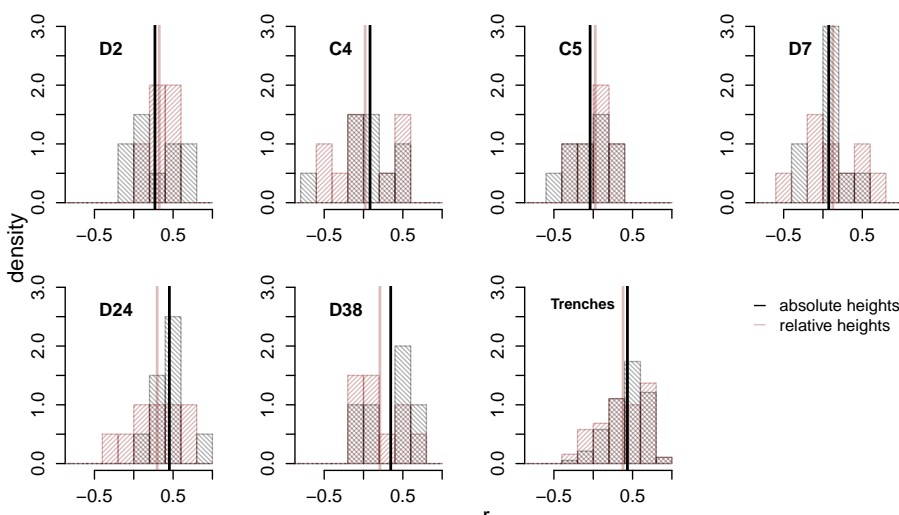

**Figure A1.** Pairwise correlation coefficients, r, of isotope profiles from different sites using absolute (black) and relative heights (red).
Vertical lines show the mean correlation coefficients.

relative heights. The difference of the mean correlations was statistically not significant for any of the sites (p < 0.05, Fig. A1).
Additionally, we were unable to identify common isotopic peaks at most sites, which also indicates that local snow heights
generally changed fast. At the same time, we were able to confirm highly irregular snow accumulations e.g., at site D38. These
findings are consistent with previous findings (e.g., Münch et al., 2016; Birnbaum et al., 2010). The snow cores in this study
are therefore processed at absolute height reference.

## Appendix B:  Slope inclination scales

We calculated slope inclinations for scales ranging from 1 to 15 km (REMA DEM, Howat et al., 2019) and tested how this af-
fected their relations with SNRs, surface roughnesses, and accumulation rates. For SNRs, the absolute values of the correlation
coefficient varied slightly for different spatial scales used to calculate slope inclinations, hereby increasing from r = - 0.82 (p <
0.05) at 1 km to r = -0.95 (p < 0.05) at 15 km (Fig. B1). A similar strengthening of the correlation with increasing spatial scale





was obtained for surface roughness and accumulation rates. Previous studies already proposed the existence of relationships

between accumulation rates, slope inclinations and snow surface features for slopes calculated at scales of 1-2 km (Eisen et al., 2005; Arcone et al., 2005; Frezzotti et al., 2002; Furukawa et al., 1996, e.g.,) and 16 km (Black and Budd, 1964). While a link between wind speed and slope inclination has been established using coarse datasets (e.g., Broeke and Lipzig, 2003; Endo and Fujiwara, 1973; Mather and Miller, 1966) this link was expected to break down with higher resolutions (e.g., 50 km, Kikuchi and Ageta, 1989). Others proposed that small slope changes at scales < 10 km and the commensurate changes in wind speed

would be able to explain differences in accumulation rates (Lenaerts et al., 2012), e.g., by sublimation (Frezzotti et al., 2004) and redistribution (King et al., 2004), as well as snow surface features (Whillans, 1975). In this study, we found that correlation coefficients between SNRs, accumulation rates, and surface roughness increased with increasing spatial scales. Yet considering the small size of our sample, the sensitivity of the correlation coefficients to different slope scales could however be dependent on a single surface undulation in the sampling area. We therefore take into account earlier results from wind simulations (e.g.,

Lenaerts et al., 2012; Whillans, 1975) and variations of accumulation rates and surface features across smaller scales (e.g., Eisen et al., 2005; Arcone et al., 2005; Furukawa et al., 1996; Frezzotti et al., 2002) and therefore do not exceed 10 km for calculating slope inclinations.

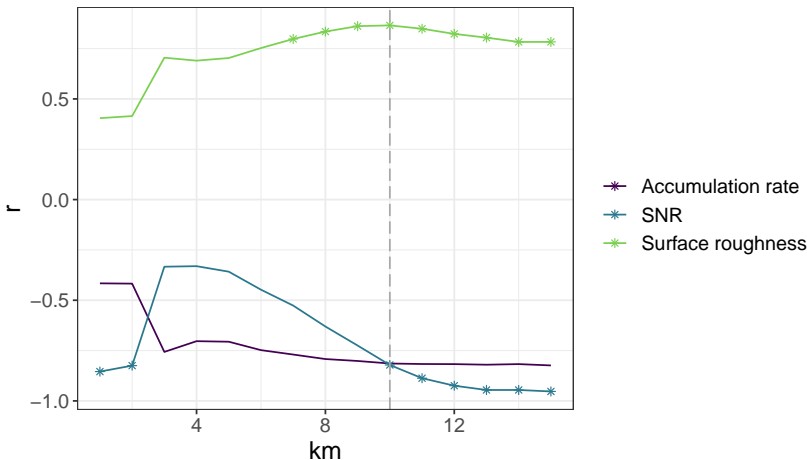

**Figure B1.** Correlation coefficients, r, between SNRs, surface roughness (SD$_{sh}$), and accumulation rates (A) with slope inclinations calculated using different scales (1-15 km). Stars indicate values with statistical significance (p < 0.05).

## Appendix C: Pairwise correlations with inter-profile spacing

Pairwise correlation coefficients for isotope profiles collected at Kohnen Station were found to increase as the interprofile

spacing drops to below 5-10 m (Münch et al., 2016), indicating that less distant isotope profiles contain more dependent noise, probably due to the spatial scales of snow surface features like sastrugi. In order quantify independent noise, we used a minimum interprofile spacing of 10 m in this study. With this setup, we did not find any relationship between spacing and



pairwise correlation coefficients (Fig. C1), which indicates that the decorrelation length of stratigraphic noise which was found to be 5 - 10 m at Kohnen station is valid for larger areas in DML.

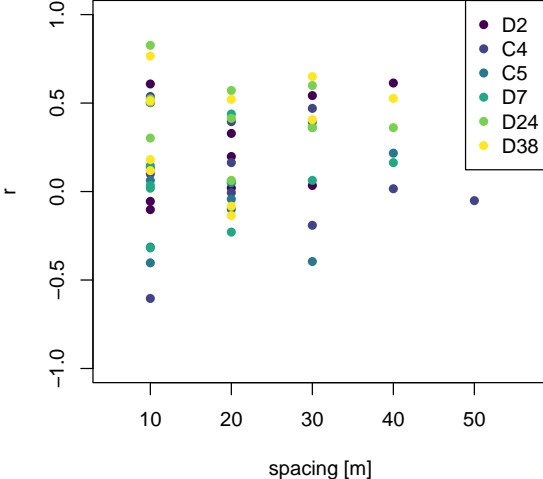

**Figure C1.** Pairwise correlation coefficients, r, of the $\delta^{18}$O profiles as a function of their interprofile spacing. Colours indicate the different sites.

*Author contributions.* TL, MH, and JF designed the expedition and TL designed the sampling strategy. NH and TL designed the study. JF, RD, TL, and MH carried out the sampling on the EAP. NH conducted the isotope measurements with the help of AZ and TM. All authors, especially TL, AZ, and TM, contributed to the scientific analysis. NH performed the analysis and wrote the manuscript, which was reviewed by all authors.

*Competing interests.* The authors declare that they have no conflict of interest.

*Acknowledgements.* We thank the scientists, technicians, and support staff at Kohnen Station for their assistance, especially Klaus Trimborn for his skilful support during sample collection. Furthermore, we would like to thank Hanno Meyer and Mikaela Weiner for their work in the isotope laboratory at AWI Potsdam, and Christoph Schneider, for the scientific supervision of the initial draft. Data analysis was performed in R: A Language and Environment for Statistical Computing. The Antarctic map is based on Quantarctica datasets in QGIS, kindly provided by the Norwegian Polar Institute (Matsuoka et al., 2018). This project received financial support from the Helmholtz foundation through the



Polar Regions and Coasts in the Changing Earth System (PACES II) programme (COMB-I project) and from the European Research Council (ERC) under the EU's Horizon 2020 Research and Innovation Programme (grant agreement no. 716092). It was furthermore supported by the Informationsinfrastrukturen Grant of the Helmholtz Association as part of the DataHub Earth and Environment.





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
