# Peer review of "Stratigraphic noise and its potential drivers across the plateau of Dronning Maud Land, East Antarctica"

_EGUsphere, 2022_

## Author Response (AR1)

**Author's Response**

Nora Hirsch[1], Alexandra Zuhr[1], Thomas Münch[1], Maria H.rhold[2], Johannes Freitag[2], Remi Dallmayr[2], and Thomas Laepple[1,3]

[1]*Alfred-Wegener-Institut, Helmholtz Centre for Polar and Marine Research, Potsdam, Germany*
[2]*Alfred-Wegener-Institut, Helmholtz Centre for Polar and Marine Research, Bremerhaven, Germany*
[3]*University of Bremen, MARUM – Centre for Marine Environmental Sciences and Faculty of Geosciences, Bremen, Germany*

We thank the reviewers as well as the editor for the helpful suggestions to improve the quality of our work. We carefully considered the comments and adjusted our manuscript accordingly. Here, we address them point by point; in a first section the ones from M. Frezzotti and in a second the ones from the anonymous reviewer. In a third section, we comment on additional changes we made to improve the study. We state changes we made in the manuscript *in green*.

1. **Reviewer 1:  M. Frezzotti**

   **The paper by Hirsch et al. deals with the use stable water isotope of snow to provide information about stratigraphic noise on ice/firn core past climate record. The main tools used in this study are the analysis of the snow stable isotope collected at trench and snow cores at 4 sites in DML. The paper goal is the improving knowledge on the ratio between climate signal versus stratigraphic noise. The manuscript subject is appropriate for Cryosphere and the subject is very important, but I don't think this paper should be published as is.**

   1.1. **Gli Autori utilizzano principalmente la stratigrafia isotopica senza altri indicatori fisico/chimici stratigrafici (ECM, DEP, nssSO4, cationi ecc.). I confronti con la stratigrafia chimica dell'elemento stabile (Na, Ca, Mg) potrebbero essere molto utili per definire l'isocrona e l'elemento chimico soggetto a riemissione come NO3 e Cl potrebbe essere indicatore di diffusione isotopica.**

      We agree that a thorough study including measurements like impurities or also densities would be interesting. We already showed differences and similarities in the stratigraphic noise of water isotopes (Münch et al., 2016, 2017) as well as densities (Laepple et al., 2016). However, this specific study is a quantification of stratigraphic noise in water isotope records across space which is relevant for using ice cores as a climate archive. Also, this study is not about diffusion itself. In our view, a more detailed assessment of diffusion is also not relevant for our conclusions, as we do not expect it to significantly differ between the cores at each site.

   1.2. **A mio parere, l'influenza principale del rapporto segnale/rumore dovrebbe essere il rapporto tra la quantità di accumulo di neve e l'entità del sastrugi, gli autori usano come rugosità superficiale la DS delle altezze superficiali. I valori SD proposti come sub decametrici non sono realistici, l'altezza dei sastrugi sull'altopiano antartico varia da decametri (20-50 cm) a 1,5 metri. Infatti, nella figura 2 gli Autori riportano altezze della neve superficiale comprese tra 20 e 40 cm, dovute alla rugosità superficiale, e gli autori hanno riportato la pendenza in metri su scala chilometrica.**

      There might be a misunderstanding - the surface topography is characterized by variations on different spatial scales. For the stratigraphic noise with a decorrelation length of several meters (Münch et al., 2016), we are interested in the local snow height variations e.g., surface

roughness, in our case on a scale of < 60 m. The slope inclinations are assessed across scales of 10 km. They are therefore two different quantities.

Furthermore, the surface roughness, as the standard deviation (SD) of the snow heights, is a statistical measure of the variation within a sample and does not give direct information on the amplitude or max/min values, e.g., sastrugi heights. Assuming a sine wave, the amplitude would be 2*√2 = 2.8 times larger than 1x SD. As we are looking at minimum and maximum values within a transect, additional stochastic variations are expected, which can result in an even higher peak to peak amplitude relative to the SD's. Therefore, the SD's of < 10 cm do not contradict the measured sastrugi heights. We still consider the SD to be the more robust metric for snow height variations related to stratigraphic noise as estimates of ranges are statistically less stable.

**1.3. Inoltre, la superficie dei sastrugi/rugosità varia continuamente durante le stagioni/anni, con ampiezza massima durante la primavera alla fine della stagione catabatica, mentre gli Autori utilizzano un'unica misurazione durante il periodo estivo (il periodo minimo di ampiezza più bassa). Gli autori hanno trascurato studi precedenti sulla variabilità locale dell'accumulo di neve dovuta alla morfologia dei sastrugi utilizzando misure di paletti (es. Mosley-Thompson et al., 1999; Ekaykin et al., 2002; Frezzotti et al., 2005, 2007; Kameda et al., 2008; Minghu et al., 2011). Questi studi hanno evidenziato la forte variabilità interannuale e la significativa variabilità spaziale a scala metrica dovuta a sastrugi e iato/erosione in accumulo, per siti con accumulo inferiore a 80-100 mm we per anno, come i siti studiati nel manoscritto proposto. Il vento tra 2 e 5 m/s è in grado di trasportare per sospensione ea velocità superiori a 5 m/s il vento è in grado di soffiare e poi sublimare la neve. Alla stazione di Kohnen si verificano per la maggior parte del tempo velocità del vento superiori a 2 m/s (Fig. 1 del manoscritto), lo sfregamento del vento e il basso accumulo sono i principali fattori di iato/erosione e la superficie vetrata con fessura termica rappresenta chiaramente questo fenomeno. Alla stazione di Kohnen, gli eventi di vento forte si verificano 10-20 volte all'anno, come determinato da Van As (2007), e producono cambiamenti nella rugosità della superficie.**

As pointed out by M. Frezzotti, the surface roughness varies throughout the year. In this study, the focus is on the relative differences in space. As all measurements were conducted in the same season, this still allows intersite comparisons.

However, the question remains whether the measurements in this single season are representative for the surface variations influencing the stratigraphic noise over time. We found that the past surface roughnesses based on isotope isochrones are similar to the surface roughness, even though they originate from different seasons. We therefore assume that the summer surface roughness is still a relevant site characteristic and could be a predictor for the overall stratigraphic noise.

We added a section in Line 203: *"The negative correlation of SNRs and surface roughnesses, thus higher undulations related to more noise, is intuitive. It indicates that, although the surface roughness represents only a snapshot of a single summer season, this snapshot is at least partly representative for the general surface roughness during the past years. If this is confirmed, the surface roughness, which is easy to measure, could be used as an indicator of the stratigraphic noise."*

We also fully agree that there are strong variations in accumulation at small spatial scales. In this study we do not neglect, but on the contrary, analyse these spatial and temporal

variations by assessing stratigraphic noise = the noise caused by local variations in accumulation and wind-drift on a scale of < 60 m.

**1.4. Nella figura 2 gli Autori correlano i minimi/massimi dell'isotopo comune nello stesso sito, ma con una differenza di circa il 10 per mille nel delta 18O che ipnotizza l'"isocrona" dello stesso evento o stagione nevosa. Questa differenza di valore non è realistica se sono "**

We agree that the assignment of isochrones in these highly variable records is affected by some degree of uncertainty - such that it was only done at two sites. At these two sites, as correctly stated by M. Frezzotti, the isotopic values representing an isochrone do not match in their value. Some of them differ in several permil - with a mean SD of 1.6 permil. These differences are in a range which we expect within at least the same season (e.g., summer or winter). Also and maybe more importantly, we do not expect the isochrones to have the same values in the snow cores, due to isotopic diffusion. The thickness of a specific layer will affect the amplitude reduction and thus lead to variations in the isotopic composition even inside isochrones.

We added an explanation on this in Line 164: *"Within these estimated isochrones, the isotope values between the snow cores show some variations. Isotope values within isochrones at location D2 and D38 have a mean SD of 1.6 ‰. Such variations can be expected within the same season (e.g., summer or winter). They also result from isotopic diffusion, as the thickness of a snow layer with a certain isotopic values will affect how much the amplitude is reduced."*

2. **Reviewer 2: Anonymous**

**This paper analyzed the stratigraphic noise across a 120 km transect on East Antarctic Plateau. At seven sites they extracted five 1 m snow cores with a 10 m interprofile spacing and measured the oxygen isotope records. The measured values were used to estimate signal to noise ratios as a measure of stratigraphic noise. The goal of this paper is to provide sampling strategies guide for high resolution isotope records.**

**This paper is well written, and the subject is appropriate for Cryosphere. However, I have still some minor comments which should be addressed before publication.**

2.1. **It would be interesting to compare the isotopes sampling with the snow properties (e.g. density or grain size) to see whether there is a connection to the stratigraphic noise. Did you also measure snow properties (like density) for each side and snow core?**

We agree that it would be interesting to compare snow densities with the isotope records, e.g., at the sites where we observed glazed surfaces. Unfortunately, we do not have any density measurements of the snow cores. Such a comparison was, however, previously carried out by Laepple et al. (2016) at the Kohnen Snow Trenches. In this study, no significant correlations were observed between isotope and density profiles, except for a relationship of the mean profiles related to a seasonal cycle in both parameters.

2.2. **I'm a bit curious about the quality of the provided results because of the low sampling numbers and locations. The author mentions two times (line 191 and 232) that the hypothesis is based on low number of samplings. Could you provide some sentences about how robust your hypothesis/assumption are and what is approx. the minimum sampling numbers and locations to make your hypothesis more robust?**

Thanks for the good suggestion. We are providing uncertainty ranges for the SNR estimates that take into account the number of samplings. For the potential relationships between SNRs and the environmental properties we state if the p-value of a correlation is below or above 0.05. We also generally recommend taking a higher number of records or longer records to make the SNR estimates more robust. Based on your suggestion we included a paragraph on how the uncertainty depends on the number and length of the cores. Within the medium range of correlation values found at the site (near gaussian distribution), we can expect that an increase of the number of snow cores or the length of cores by factor n will reduce the standard error of the correlations by $1/\sqrt{n}$. The corresponding new text is added in Line 241: *"In a first order approximation, an increase of the number of snow cores or the length of cores by factor n will reduce the standard error of the correlations by $1/\sqrt{n}$. That means, increasing the number of cores by factor 4, already decreases the uncertainty of the pairwise correlations by half."*

To avoid the costs and workload which comes with replicate isotope profiles if aiming for lower uncertainties, we therefore also propose other methods like radar to make the hypothesis on the proposed relationship more robust.

2.3. **Line 69-70: Could you elaborate a bit more why you only took one line profiles perpendicular to the dominant large scale wind direction and not more lines parallel to the large scale wind direction? I would expect that also having profiles parallel to the wind field would help to show that the impact from other wind direction can be excluded.**

We agree with the reviewer, that ideally, taking profiles at different directions relative to the wind would be useful to study the influence of the wind direction.

Here, time constraints in the expedition didn't allow to systematically study this aspect and we therefore decided to use the same sampling direction as we used in earlier studies on stratigraphic noise (Münch et al., 2016, Laepple et al., 2016, Münch et al., 2017) to allow a comparison. Across the region covered in this study, including the area around Kohnen Station, the main wind direction is similar.

Furthermore, snow dunes form parallel to the main wind direction. Thus, in order to measure the same amount of dunes, a shorter measuring distance is required if the measurements are taken perpendicular to the wind direction compared to if they are taken parallel to the wind direction.

We added this to the revised version (Lines 70 – 73): *"The directionality was chosen to allow a comparison to the studies at the Kohnen trenches (Münch et al., 2016, 2017). As the snow dunes in the study region are predominantly parallel to the wind direction, measuring perpendicular to the wind ensures to better sample the dunes along the 60 m overall distance."*

2.4. **Figure 2: Could you provide some information whether you corrected your snow core data? Does the isotope signal correspond to the actual snow depth?**

The depth in the snow-profiles correspond to the actual snow depth. The method of using thin carbon tubes for sampling is well established (e.g. Schaller et al., 2016). In some cases, in the process of pushing the tube into the snow, extracting the snow core, cutting and

handling, minor errors can occur, which is while we expect the depth scale uncertainty to be < 2 cm.

We have such a description in the method section, which we refined and elaborated: "Compression or expansion during handling, transporting and cutting the snow cores resulted in a maximum depth uncertainty of 2 cm and slight variations (41-43) in the number of samples per profile. Combined with the maximum uncertainty of 1 cm resulting from the snow height measurements (Sect. 2.6), the absolute depths values have a combined maximum uncertainty of 3 cm." (Lines 76 – 79).

2.5. **Line 215: Could you elaborate this more? What kind of signal could the D24 isotope profiles represent instead?**

At location D24, the isotope profiles are rather constant through depth (with a low variance) except for two layers (one depleted and one enriched) that appear as strong anomalies at the same depth in all cores. These 'flat' isochrones contrast the high surface roughness at D24 (that is strongest across all sites). Further, the profiles are not consistent to the expectation of a continuous preserved climate record that should contain 5-7 seasonal cycles. This anomalous behavior is also reflected at the surface: here, we observe strong spatial variability with a mix of glazed snow fields and patches characterized by high sastrugi - a feature that was (visually) unique at this site compared to the other sites.

We thus speculate that D24 represents a discontinuous record in which the two anomalies / isochrones probably represent a summer and a winter layer from any time within the 5-7 years of accumulation.

The SNR defined in this study is the ratio of the signal shared between the cores and the local variations; therefore, the estimated SNR is high for this site, driven by the two anomalies. However, if our interpretation is right, the record could not be used for seasonally to annually resolved climate reconstructions.

We refined our text in the following way (Lines 214 – 220): "We speculate that D24 represents a discontinuous record in which the two flat anomalies / isochrones represent a summer and a winter layer from any time within the nearly 7 years of accumulation. The SNR, as defined in this study, is the ratio of the shared signal between the records and the local variations. Therefore, the estimated SNR at site D24 is high, driven by the two anomalies. Between the two anomalies, the past surface roughnesses could have been as high as the present surface roughness. We therefore assume that the spatially coherent part at this site does not represent a seasonal or annual climatic signal and that the isotope records do therefore not provide any useful information for climate signal interpretations."

2.6. **Minor comments**

**Line 101: Could you provide information about what kind of interpolation you used? Linear interpolation?**

Yes, we used linear interpolation. As the isotope profiles are smooth on the cm scale (due to isotopic diffusion), the details of the interpolation method do not affect our results. The confidence intervals and significance values are also not affected as they are based on bootstrapping (and do not rely on the number of datapoints that is artificially increased by the interpolation). We added the method in the revised text in Line 105.

**Figure 3 caption: what is several km?**

We added the km distance from the locations to the transect in the caption.

3. **General comment from the authors**

To underline the significance of the results, we will include a discussion section on the relationship between the SNRs and the ability to recover a climate signal at a certain time resolution; an isotope dataset with an increased SNR will result in the recovery of a climate signal of a higher resolution, which we expect can be achieved by using our sampling recommendations.

We added: "We suggest that an optimal site selection might allow to increase the effective resolution of climate reconstructions from the East Antarctic Plateau. As the stratigraphic noise considerably varies over relatively small spatial scales (~100km), the optimal choice of a drilling location can minimise the stratigraphic noise and thus improve the SNRs within snow or firn cores. In most cases, the SNR and not the measurement resolution is the limiting factor for the temporal resolution of the climate signal that can be recovered (Münch and Laepple, 2018). The higher the noise level, the more averaging in time is needed to reduce the uncorrelated (white) noise while preserving the more persistent (red) climate signal. As an example, when assuming a climate signal with a Power Spectral Density = f−beta, with beta = 1, and uncorrelated noise, a reduction of the noise by a factor of n would increase the obtainable climate resolution by the same factor. Assuming that 50 % of the noise is stratigraphic noise (Laepple et al., 2018), we would expect up to four times the obtainable climate resolution at Kohnen (SNR of 0.77) relative to C4 (SNR = 0.1)." (Lines 256 – 265)

We furthermore reacted to the comments by refining the descriptions of data and methods by minor adjustments, so that it is better understandable.

**Literature**

Laepple, T., M. Hörhold, T. Münch, J. Freitag, A. Wegner, and S. Kipfstuhl (2016). Layering of surface snow and firn at Kohnen Station, Antarctica: Noise or seasonal signal?. J. Geophys. Res. Earth Surf., 121, doi:10.1002/2016JF003919.

Münch, T., Kipfstuhl, S., Freitag, J., Meyer, H., & Laepple, T. (2016). Regional climate signal vs. local noise: a two-dimensional view of water isotopes in Antarctic firn at Kohnen Station, Dronning Maud Land. *Climate of the Past*, *12*(7), 1565-1581.

Münch, T., Kipfstuhl, S., Freitag, J., Meyer, H., & Laepple, T. (2017). Constraints on post-depositional isotope modifications in East Antarctic firn from analysing temporal changes of isotope profiles. *The Cryosphere*, *11*(5), 2175-2188.

Schaller, C. F., Freitag, J., Kipfstuhl, S., Laepple, T., Steen-Larsen, H. C., & Eisen, O. (2016). A representative density profile of the North Greenland snowpack. The Cryosphere, 10(5), 1991–2002. https://doi.org/10.5194/tc-10-1991-2016

---

## Referee Report (RR1)

**Stratigraphic noise and its potential drivers across the plateau of Dronning Maud Land, East Antarctica**

The author addressed my comments well and improved the manuscript. I have two minor suggestion which should be included in the discussion of the manuscript before publication.

- Would it be possible to provide a paragraph with the 'best practice' procedure for next campaigns to provide solid data to better explain the stratigraphic noise in order to increase the effective resolution of climate reconstructions from the East Antarctic Plateau. Like what type of measurements should be included/or not (e.g. isotope, snow density, snow SSA, precipitation patterns, depositional conditions, accumulation rate, slope inclination, surface roughness, direction/distances/size of the locations etc.) and the why?
- Could you elaborate a bit whether snow metamorphism and the vapor exchange between snow and atmosphere due to sublimation and/or deposition could also have an impact on the stratigraphic noise?

Minor comments:

- Table 2: The 'O' in d18O is sometimes italic, sometimes not. Not consistent with the text.

---

## Author Response (AR2)

**Author's Response #2**

**Nora Hirsch1, Alexandra Zuhr1, Thomas Münch1, Maria Hörhold2, Johannes Freitag2, Remi Dallmayr2, and Thomas Laepple1,3**

1Alfred-Wegener-Institut, Helmholtz Centre for Polar and Marine Research, Potsdam, Germany 2Alfred-Wegener-Institut, Helmholtz Centre for Polar and Marine Research, Bremerhaven, Germany 3University of Bremen, MARUM – Centre for Marine Environmental Sciences and Faculty of Geosciences, Bremen, Germany

We thank the editor, the anonymous reviewer #2, as well as the new anonymous reviewer #3 for taking the time to help to improve our work. We adjusted our manuscript accordingly. Additionally, we again revised and improved the language and adjusted some minor details to make the manuscript easier to read and more understandable. Here, we address the reviewer comments point by point, stating the according changes we made in the manuscript *in green*.

**1. Anonymous referee #2 (Report 1):**

The author addressed my comments well and improved the manuscript. I have two minor suggestion which should be included in the discussion of the manuscript before publication.

1.1 Would it be possible to provide a paragraph with the 'best practice' procedure for next campaigns to provide solid data to better explain the stratigraphic noise in order to increase the effective resolution of climate reconstructions from the East Antarctic Plateau. Like what type of measurements should be included/or not (e.g. isotope, snow density, snow SSA, precipitation patterns, depositional conditions, accumulation rate, slope inclination, surface roughness, direction/distances/size of the locations etc.) and the why?

Thanks for this suggestion. We agree with the reviewer that it would be interesting to study additional processes that influence the isotopic composition in order to deepen our understanding of the snow deposition and the signal formation. We already mention the possibility of detailed surface observations as a helpful extension to our study (Zuhr et al., 2021, Picard et al., 2019). Densities and SSA measurements could give some indications regarding the stratigraphy, however, the relationship of both densities as well as SSA to stable water isotopes is not unambiguous and not well understood (Laepple et al., 2016, Stuart et al., 2023). They are therefore not suggested in our study.

So far, we have a section on implications for future studies (Sect. 4.5), which includes suggestions for optimized sampling locations to achieve a high resolution climate signal from the EAP. However, based on your comment, we will clearly distinguish our different implications and move part of the text to a new section 4.6 named: "Suggestions for optimal site selection for high resolution climate reconstructions from the late Holocene".

We divide this new section into three parts: first, we talk about the importance and possible strategies for site selection as before. Then, we suggest a best practice

sampling setup, which was also used in this study, by adding the following sentences (Lines 283):

"The sampling setup at the thoroughly selected sites should follow the suggestions of Münch et al. (2016): the distance of replicate cores should be larger than the expected decorrelation length of stratigraphic noise, for example 10 m in the DML plateau area. The number of cores should be chosen based on the expected amount of stratigraphic noise and the intended signal resolution. Based on the findings by Münch et al. (2016), we suggest to take 5 replicates at locations with similar environmental properties to Kohnen Station. The sample direction should be perpendicular to the overall wind direction if the surface roughness is measured across the sampled cores as in this study."

Finally, we add a paragraph about the signal interpretation regarding sublimation/snow metamorphism and precipitation intermittency: While the quantitative impact of sublimation/snow metamorphism on the isotopic composition is still a matter of debate (Wahl et al., 2022), it has been shown that precipitation intermittency might be responsible for 50 % of the total noise variance in local isotope records (Laepple et al., 2018, Münch et al., 2021). These two processes are probably coherent across our study region but should in general be considered when interpreting isotope records. We therefore add (Line 289):

"Signal interpretation should further consider influences on the isotopic composition from, e.g., sublimation (Wahl et al. 2021), snow metamorphism (Stuart et al. 2023) and precipitation intermittency. The latter can be responsible for up to 50 % of the noise variance across large spatial scales (Laepple et al., 2018). The sampling strategy we propose here could therefore be expanded by replicate cores taken at optimal distances to account for precipitation intermittency, as suggested by Münch et al., (2021)."

**1.2 Could you elaborate a bit whether snow metamorphism and the vapor exchange between snow and atmosphere due to sublimation and/or deposition could also have an impact on the stratigraphic noise?**

Stable water isotopologues can be altered when exposed to the atmosphere, before or after deposition (Wahl et al. 2021, Stuart et al. 2023, Ebner et al. 2017). Both snow metamorphism and sublimation do however, not change the stratigraphy itself, and do therefore not notebly influence stratigraphic noise. Instead, they can introduce an overall isotope bias and need to be accounted for in the overall isotope interpretation (Wahl et al. 2022). As elaborated in comment 1.1, we now mention the need for a cautious signal interpretation that also considers sublimation and snow metamorphism in a new section 4.6.

**1.3 Minor comments: Table 2: The 'O' in d18O is sometimes italic, sometimes not. Not consistent with the text.**

Thank you. We corrected the style of the O in  $\delta_{18}O$  to match the style of the text and figures.

2. Anonymous referee #3 (Report 2):

In their study, the authors quantify the relationship between stratigraphic noise in the stable water isotopic signal of the top 1 m snow and the environmental properties inclination, surface roughness, and accumulation rate. Based on these findings, the authors aim to provide guidance for future snow, firn, and ice core drillings to obtain higher SNR. The paper is well-structured and presented in a clear way. The impact of the results is discussed appropriately within the limitations of the relatively low number of analyzed snow cores, and given p-values allow for reasonable assessment. The study provides useful contributions to the assessment of stratigraphic noise in Antarctic snow records and will be an appropriate contribution to The Cryosphere after some minor revisions:

2.1 Figure 5: The study, particularly Figure 5, would benefit from clarity on the uncertainties in the environmental properties. I recommend adding horizontal uncertainty bars in Figure 5, but at least the uncertainties for surface roughness, accumulation rate, and inclination should be provided in the discussion.

Thank you for this very good suggestion. We added the following method description in section 2.6:

Uncertainty of the slope inclinations, Line 117: "To assess the uncertainty of these estimates, we calculate the slope inclinations with the same azimuth over 10 km segments across 36 different points, located at 200, 400 and 600 m around each study site (12 different directions in steps of 30°) and extract the SD of these slope inclinations."

Uncertainty of accumulation rates, Line 122: "To get an estimate for the uncertainty of these values, we use the accumulation rate over the last 200 years from the B32/DML05 ice core at Kohnen Station (Oerter et al. 2000). We calculate the SD of the 5-year black averaged record, since 5 years roughly represents the accumulation period in our snow cores. For each site, we scale the SD to the local mean accumulation rate.

Uncertainty of surface roughnesses, Line 128: "Further, we resample with replacement the height values from each site 1000 times, estimate the surface roughness from these samples, and use the SD of these surface roughness values as a measure of uncertainty."

We added these uncertainty estimates as horizontal error bars to manuscript Figure 5 (see Fig. 1 below) and also to the new figure in Appendix D (see your comment 2.3 and Fig. 2 below).

• sastrugi surface (D2, C4, C5, D7, D38, Trenches)

**2.2 L197 (lines in initially submitted manuscript): add p-value for correlation SNR with accumulation rates, and L204: p-value for r?**

The p-values were already added in the second version of the manuscript. Instead of only indicating if a value is smaller than 0.05 (in the first version of the manuscript), we now state the according p-value for each correlation.

**2.3 L219: add figures in the style of Figure 5 to Appendix to support the values presented in 4.4**

We appreciate this good suggestion and added an according figure in Appendix D (Fig.2 below).

• mix of sastrugi and glazed surface (D24)

Figure 1: Comparison of signal to noise ratios (SNR) to accumulation rate A [mm w.e. a-1], surface roughness SDSH [cm], and slope inclination [m km-1]. Most sites were dominated by sastrugi while at D24 we observed a mix of sastrugi and glazed surfaces. The latter was therefore excluded in the linear regression analysis (dashed lines). Vertical lines indicate the 95% confidence intervals of the SNR estimates, while horizontal lines represent the uncertainty of the environmental properties (2 \* SD). Uncertainty of 10 km slope inclinations are very small such that they are not visible for most of the sites.

Figure 2: Comparisons between accumulation rate A [mm w.e.  $a_{-1}$ ], surface roughness SDSH [cm], and slope inclination [m km-1]. Vertical and horizontal lines represent 2 \* SDs of the according environmental property as an indication for uncertainty. Linear regression lines (dashed) suggest possible relationships. Uncertainty of 10 km slope inclinations are very small such that they are not visible for most of the sites.

**Literature**

Ebner, P.P., Steen-Larsen, H.C., Stenni, B., Schneebeli, M. and Steinfeld, A., (2017). Experimental observation of transient  $\delta$  18 O interaction between snow and advective airflow under various temperature gradient conditions. *The Cryosphere*, *11*(4), pp.1733-1743.

Laepple, T., M. Hörhold, T. Münch, J. Freitag, A. Wegner, and S. Kipfstuhl (2016). Layering of surface snow and firn at Kohnen Station, Antarctica: Noise or seasonal signal?, J. Geophys. Res. Earth Surf., 121, doi:10.1002/2016JF003919.

Münch, T., Werner, M., & Laepple, T. (2021). How precipitation intermittency sets an optimal sampling distance for temperature reconstructions from Antarctic ice cores. *Climate of the Past*, *17*(4), 1587-1605.

Oerter, H., Wilhelms, F., Jung-Rothenhäusler, F., Göktas, F., Miller, H., Graf, W., and Sommer, S. (2000). Accumulation rates in Dronning Maud Land, Antarctica, as revealed by dielectric-profiling measurements of shallow firn cores, Annals of Glaciology, 30, 27–34.

Picard, G., Arnaud, L., Caneill, R., Lefebvre, E., and Lamare, M. (2019). Observation of the process of snow accumulation on the Antarctic Plateau by time lapse laser scanning, The Cryosphere, 13, 1983 – 1999, https://doi.org/https://doi.org/10.5194/tc-13-1983-2019.

Stuart, R.H., Faber, A.K., Wahl, S., Hörhold, M., Kipfstuhl, S., Vasskog, K., Behrens, M., Zuhr, A. and Steen-Larsen, H.C., (2023). Exploring the role of snow metamorphism on the isotopic composition of the surface snow at EastGRIP. *Cryosphere Discussions*.

Wahl, S., Steen-Larsen, H. C., Reuder, J., and Hörhold, M. (2021). Quantifying the Stable Water Isotopologue Exchange Between the Snow Surface and Lower Atmosphere by Direct Flux Measurements, Journal of Geophysical Research: Atmospheres, 126, e2020JD034 400, 510 https://doi.org/10.1029/2020JD034400

Wahl, S., Steen-Larsen, H.C., Hughes, A.G., Dietrich, L.J., Zuhr, A., Behrens, M., Faber, A.K. and Hörhold, M., (2022). Atmosphere-Snow Exchange Explains Surface Snow Isotope Variability. *Geophysical Research Letters*, *49*(20), p.e2022GL099529.

Zuhr, A. M., Münch, T., Steen-Larsen, H. C., Hörhold, M., and Laepple, T. (2021). Local-scale deposition of surface snow on the Greenland ice sheet, The Cryosphere, 15, 4873–4900, https://doi.org/10.5194/tc-15-4873-2021, publisher: Copernicus GmbH.